# Knockouts of Yeast Plasma Membrane Phosphate Transporters Alter Resistance to Heavy Metals

Larisa Ledova [1], Lubov Ryazanova [1], Ludmila Trilisenko [1], Vladimir Ostroumov [2] and Tatiana Kulakovskaya [1,*]

[1] Skryabin Institute of Biochemistry and Physiology of Microorganisms, Federal Research Center "Pushchino Scientific Center for Biological Research of the Russian Academy of Sciences", Pushchino 142290, Russia; tril2020tril@rambler.ru (L.T.)

[2] Institute of Physicochemical and Biological Problems of Soil Science, Federal Research Center "Pushchino Scientific Center for Biological Research of the Russian Academy of Sciences", Pushchino 142290, Russia

* Correspondence: alla@ibpm.ru

## Abstract

Mutant yeast strains with altered sensitivity to heavy metals are crucial for revealing the mechanisms of metal absorption and detoxification, as well as for bioremediation of these pollutants. Here, we show that a knockout of the *PHO87* gene encoding the low-affinity phosphate transporter of the cytoplasmic membrane of *S. cerevisiae* increased resistance to manganese, silver, and vanadate ions. However, a knockout of *PHO90* (*PHO87* paralog) did not affect the sensitivity to silver and vanadate ions but increased sensitivity to manganese ions. The Δ*pho87* cells accumulated 10 times less manganese compared to the wild-type cells, while the Δ*pho90* cells accumulated two times more manganese compared to the wild-type cells, when grown in YPD with 2 mM $MnSO_4$. The polyphosphate content of the Δ*pho84*, Δ*pho87*, and Δ*pho90* cells cultivated at high phosphate concentration did not differ from that of the wild-type strain. In the presence of 2 mM $MnSO_4$, Δ*pho87* cells contained several times less polyphosphates, and Δ*pho90* cells contained more short-chain polyphosphates than the cells of the wild-type strain. We hypothesize that phosphate carriers participate in the regulation of heavy metal uptake, and the respective knockouts are useful in bioremediation and bioassay of these pollutants.

**Keywords:** *PHO87*; *PHO90*; phosphate transporter; yeast; manganese; silver; vanadate; polyphosphate

## 1. Introduction

Pollution of water and soil with heavy metal ions is an important problem in environmental protection because these pollutants can lead to a variety of health problems in humans, including cancer and neurodegenerative diseases [1]. Heavy metal ions cause epigenetic changes harmful to future generations, such as abnormal DNA methylation, aberrant histone modifications, and altered expression of non-coding RNAs [1]. One of the biotechnological approaches to soil and water remediation from heavy metals is biosorption using a variety of microorganisms [2], from bacteria [3] to yeast [4,5] and algae [6].

Several reviews have been devoted to the problem of using yeast to purify wastewater from heavy metals [5,7,8]. Various yeast species are among the suitable microorganisms for the development of biosorption technologies, since they are easily cultivated, produce a high yield of biomass, have high viability under adverse conditions, and are capable of accumulating ions of common metal pollutants [8]. The high ability of yeast to adsorb

heavy metals in non-sterile and inexpensive conditions is also the reason for the interest in the practical application of these microorganisms.

Using yeast, effective, biocompatible, and reusable hybrid nanomaterials can be obtained, including chitosan and yeast nanofibers, nanomats, nanopaper, biosilicon hybrids, and TiO$_2$ and yeast nanocomposites [7]. In the context of depollution, these nanohybrid materials can be used to improve the stability and functionality of the devices for wastewater treatment [7].

The genetic methods allow new strains to be created with unique changes in the cell surface structure and increased capability of absorbing heavy metal ions. By overexpressing metal transporters and genetically altering metal trafficking pathways, yeasts have been engineered to absorb chromium, arsenic, and cadmium in quantities 10–100 times greater than wild-type strains [9]. Moreover, new strains have been constructed that are capable of selective removal of cadmium or strontium [9]. Among practical approaches is the construction of strains overexpressing metal-binding hexapeptides (MeBHxPs) [10].

Along with genetically engineered strains, it is also important to search for new species and isolates that might improve absorption capacity for metal ions. For example, the *Geotrichum* sp. yeast isolated from polluted mangrove soils absorbs Zn$^{2+}$ and Ni$^{2+}$ into the cell and Cu$^{2+}$ into the cell wall [11]. A newly discovered marine strain of *Yarrowia lipolytica* had increased resistance to Pb$^{2+}$, Cr$^{3+}$, Zn$^{2+}$, Cu$^{2+}$, As$^{5-}$, and Ni$^{2+}$ [12]. *Yarrowia phangngaensis* X1 strain showed high levels of copper, zinc, and cadmium uptake [13]. The *Cryptococcus humicola* yeast is resistant to many metal ions; for example, it grows at 10 mM Mn$^{2+}$, which is lethal for commonly studied yeast species such as *S. cerevisiae* [14].

Yeasts are of interest not only as a material for bioadsorption devices and filters, but also for biomonitoring the presence of heavy metals in wastewater. Strains with increased resistance and absorption capacity are important for the first task, and those with high sensitivity to heavy metal ions are useful for the second. A method of bioassay of arsenate and chromium ions was proposed by using a standard estimation of *S. cerevisiae* growth inhibition [15].

Mechanisms of heavy metal uptake and detoxification in yeast are diverse and include transport proteins of the cytoplasmic and intracellular membranes, metal-binding peptides, glutathione-dependent proteins, and vacuolar compartmentalization and sequestration [4,16–18]. In addition, a significant role is played by proteins responsible for homeostasis of mineral phosphorus compounds. In yeast, the proteins of the PHO pathway play a key role in phosphate uptake homeostasis and participate in the overall regulation of yeast metabolism [19–23]. Both phosphate (Pi) [24] and inorganic polyphosphate (polyP) [25] are capable of forming complexes with heavy metals, thereby reducing their toxicity. Therefore, phosphate transporters may be involved in the heavy metal detoxification mechanisms. The cytoplasmic membrane of *S. cerevisiae* possesses four phosphate transporters with different properties: the two low-affinity H+/Pi symporters, Pho87 and Pho90, the high-affinity H+/Pi symporter Pho84, and the high-affinity Na+/Pi symporter Pho89 [19–23]. These transporters are characterized not only by different affinities to Pi but also by diverse regulatory properties [26]. For example, the expression of the *PHO87* and *PHO90* genes is independent of Pi concentration in the medium [26]. The transcription of *PHO84* and *PHO89* is induced under Pi limitation [26].

With regard to the role in metal ion homeostasis, the best-studied transporter is Pno84, which has been known as a phosphate carrier for a long time [27]. It is also a low-affinity manganese transporter, which is also capable of transporting cobalt, zinc, and copper ions [28]. Besides, the cells of *S. cerevisiae* overexpressing this transporter demonstrated enhanced arsenate uptake [29]. The multiple roles of this carrier in membrane transport and Pi sensing in yeast have been noted in the literature [30,31]. Of particular interest are Pho87

and Pho90: they are encoded by paralogous genes, but exhibit functional differences [26,32]. This has been clearly shown using the knockout of *PHO4*, which encodes the transcription factor Pho4 that activates the expression of genes of phosphate starvation response under Pi limitation. In Δ*pho4* cells, rapid restoration of Pi transport and growth at phosphate limitation is provided by the Pho87 transporter, but not by Pho90 [32]. The conditions of encocytosis into the vacuole are also different for these proteins [32]. Until now, there has been no data reported regarding the participation of Pho87, Pho89, or Pho90 in the heavy metal resistance mechanisms.

The aim of this study was to compare *S. cerevisiae* knockouts of plasma membrane Pi transporter genes *PHO87*, *PHO90*, and *PHO89* in terms of resistance to silver, vanadate, and manganese.

## 2. Materials and Methods

### 2.1. Yeast Strains and Cultivation

The strain BY4741 (MATa his3Δ1 leu2Δ0 lys2Δ0ura3Δ0) as a parent strain (wt) and its derivative knock-out strains Δ*pho84*, Δ*pho87*, Δ*pho89*, and Δ*pho90* were obtained from the collection (https://horizondiscovery.com/en/non-mammalian-research-tools/yeast, accessed on 3 March 2019).

The cultures were maintained on a YPD agar medium and cultivated in 100 mL of YPD medium containing 1% yeast extract (Sigma-Aldrich, Saint Luis, MI, USA), 2% peptone (DIAM, Moscow, Russia) and 2% glucose in Erlenmeyer flasks at 28 °C and 145 rpm for 24 h. The medium contained 3.5 mM Pi.

### 2.2. Sensitivity to Manganese and Vanadate Ions

Yeast cell culture samples (final cell concentration $0.5 \cdot 10^7$ cell/mL) were added to the wells of sterile plates containing 0.2 mL YPD medium supplemented with appropriate concentrations of $MnSO_4 \cdot H_2O$ (analytical grade, Reachim, Moscow, Russia). The cells were cultured at 30 °C and 700 rpm for 3 days, and the absorption was measured at 594 nm with a microplate photometer (MZ Sapphire, Moscow, Russia).

In the case of assay of the effects of vanadate on yeast growth, the cells of strains under study were cultivated in the wells of sterile plates containing 200 mL Pi-limited YPD medium supplemented with appropriate concentrations of $Na_3VO_4 \cdot 12 H_2O$ (analytical grade, DIAM, Moscow, Russia). The initial cell concentration in wells was $1 \cdot 10^7$ cells/mL. The Pi-limited culture medium contained 0.08 mM Pi and was prepared as described [33]. The cells were cultured at 30 °C and 700 rpm for 1 day, and the absorption was measured at 594 nm using a microplate photometer (MZ Sapphire, Moscow, Russia).

### 2.3. The Treatment of the Cells with $Ag^+$

The cells of strains under study grown in YPD medium and harvested as described above were used for treatment with $AgNO_3$ (analytical grade, Diakon, Pushchino, Russia). The cell samples of 25 mg of standardized wet biomass were placed in 0.5 mL MiliQ water with or without $AgNO_3$ and incubated for 30 min at 30 °C and stirring. After that, the cells were removed by centrifugation at $7000 \times g$ for 3 min, and the released Pi was determined in the supernatants by a malachite green colorimetric assay [34].

### 2.4. Polyphosphate Assay

For the extraction and estimation of Pi and polyP quantity, the cells were cultivated in Erlenmeyer flasks at 28 °C and 145 rpm in control YPD and in YPD supplemented with 2 mM $Mn^{2+}$. Biomass samples for polyP extraction were obtained by cultivation for 24 h in the control medium (stationary phase). In the medium with manganese, the stationary

growth phase was reached in various periods for strains under study: 4 days for the wild type (wt) strain and 3, 3, and 6 days for the strains Δ*pho87*, Δ*pho84* and Δ*pho90*, respectively.

The cells were separated from the medium by centrifugation at $5000\times g$ for 15 min and washed twice with distilled water under the same centrifugation conditions. Wet biomass samples, 150 mg each, were frozen, stored at $-20$ °C, and used for polyP extraction and assay. Pi, acid-soluble polyP fraction (polyP1), and acid-insoluble polyP fraction (polyP2) were extracted at 0 °C as described previously [35]. The acid-soluble polyP1 was extracted from biomass samples by the treatment with 0.5 M $HClO_4$ at 0 °C for 20 min under continuous stirring. After that, the samples were centrifuged at $7000\times g$ for 3 min, and the supernatants were collected. The extraction was repeated twice. The combined supernatants contained acid-soluble polyP1. The content of Pi before hydrolysis and after polyP hydrolysis in 1 M HCl for 20 min at 90 °C [35] was determined. The difference between these values was the content of acid-soluble polyP1 expressed as Pi [35]. The remaining biomass was treated with 1M $HClO_4$ at 100 °C for 10 min. After cooling, the samples were centrifuged at $7000\times g$ for 3 min, and the content of acid-insoluble polyP2 was assayed by the Pi quantification. Pi was quantified by a malachite green colorimetric assay [34] with a microplate photometer (MZ Sapphire, Moscow, Russia). To compare the polyP content, we used the quantities of polyP as Pi normalized to sample masses (μmol P/g of wet biomass). The wet biomass samples were standardized using the same centrifugation conditions, $5000\times g$ for 15 min, as in [35].

### 2.5. Manganese Assay

The biomass samples from the cultures grown in the presence of 2 mM $Mn^{2+}$ were obtained using the same cultivation conditions and time of cultivation as for the polyP assay. Biomass samples were obtained by cultivation for 24 h in control YPD. In the YPD with manganese, the stationary growth phase was reached in various periods for strains under study: 4 days for the wt strain and 3, 3, and 6 days for the strains Δ*pho87*, Δ*pho84*, and Δ*pho90*, respectively. To compare the manganese content, we used the quantities of manganese normalized to sample masses (μmol Mn/g of wet biomass). The wet biomass samples were standardized using the same centrifugation conditions, $5000\times g$ for 15 min, as in [35].

The samples of 25 mg wet biomass were burned at 150 °C in 0.2 mL 65% $HClO_4$, and manganese was assayed by atomic absorption spectroscopy.

### 2.6. Statistics

The experiments were performed in 3–4 replicates; the number of replicas is indicated in the figure captions. All figures show average values from the indicated number of experiments and the whiskers denote standard deviation. The statistical significance was assessed with the two-tailed Student's *T*-test using standard Excel software and was performed for data for mutant strains studied against data for the wt strain.

## 3. Results

### 3.1. The Effect of Vanadate on Yeast Growth

Vanadate inhibits the growth of yeast cells [36]. In this regard, vanadate compounds have been proposed as fungicides; for example, the β-$AgVO_3$ gel showed physicochemical stability and antifungal activity [37]. Vanadate ions inhibit the activity of plasma membrane proton ATPase [38]. Phosphate is a competitive inhibitor of vanadate uptake, and vanadate is a competitive inhibitor of phosphate uptake [39]. It is known that in *O. parapolymorpha*, inactivation of the *PHO87* gene leads to increased resistance to vanadate [40]. In this

regard, we tested the effects of knockouts of the genes encoding phosphate transporters of *S. cerevisiae* on yeast growth in the presence of different concentrations of vanadate.

When using YPD medium with a relatively high phosphate concentration (3.5 mM), the effect of vanadate on growth was insufficient, and the difference in growth between the strains studied could not be assessed. Therefore, we used YPD medium with a reduced phosphate concentration. All strains except Δ*pho84* grew similarly on the YPD with 0.08 mM Pi and on the YPD with 3.5 mM Pi (see the data for control growth in Figure 1). The cells of Δ*pho84* grew significantly worse even in the control, and the concentrations of vanadate used suppressed the growth of this strain almost completely. The cells of the strain Δ*pho90* showed no difference from the wt strain, while the cells of mutants Δ*pho87* and Δ*pho89* showed greater resistance compared to the wt strain.

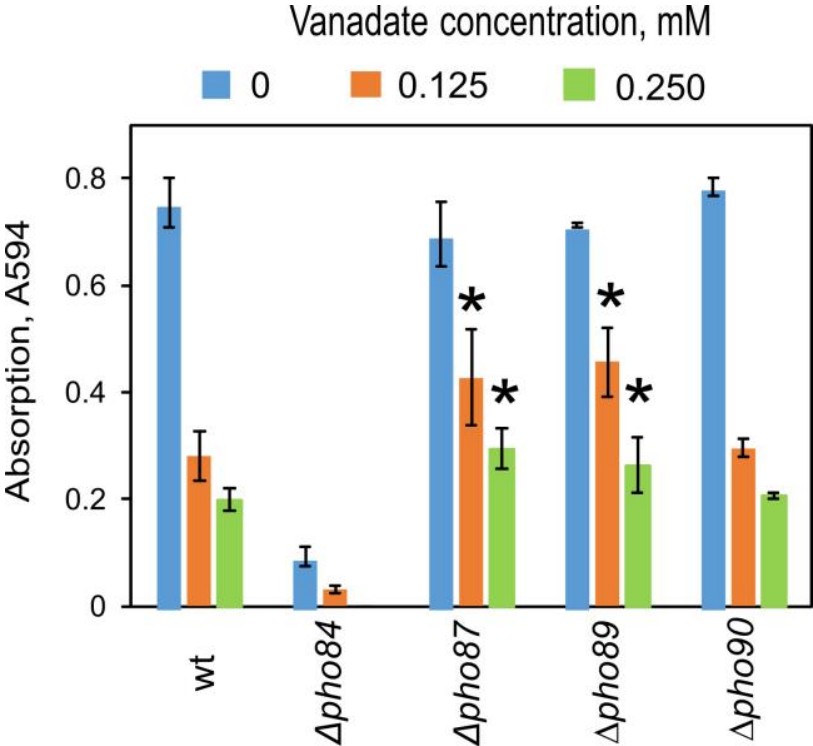

**Figure 1.** The effect of vanadate on the culture absorption of *S. cerevisiae* strains. The cells were cultured at 30 °C and 700 rpm for 1 day in the YPD medium with 0.08 mM Pi. The experiments were performed in four replicates; the values denote mean and the whiskers denote s.d. * $p < 0.01$; no designation—n.s.; two-tailed Student's *T*-test against wt.

### 3.2. The Effect of Ag⁺ on Pi Release

Silver ions and nanoparticles containing silver ions are considered promising antimicrobial agents [39]. Nanoparticles containing silver release silver ions, which directly affect microbial cells [40]. The mechanism of action of this ion is associated with damage to the cytoplasmic membrane and the appearance of an excess concentration of ROS [41]. The effect of silver ions on yeast cells leads to the release of potassium ions [42] and phosphate [43] from the cells. YPD medium does not allow us to assess the effect of silver ions on growth, since a precipitate forms in this medium. Therefore, to assess the effect of silver ions on mutants, we used a previously developed approach that assesses membrane damage by the release of Pi from cells [44].

The concentration range of AgNO₃ was 25–200 µM. The result is shown in Figure 2: the Pi yield in the solution after incubation of yeast cells in the presence of various concentrations of Ag⁺ was significantly reduced for mutant strains Δ*pho87* and Δ*pho89* compared

to the wt strain, and Δ*pho84,* and Δ*pho90* strains. This indicates the enhanced resistance of the cells of Δ*pho87* and Δ*pho89* strains to Ag$^+$.

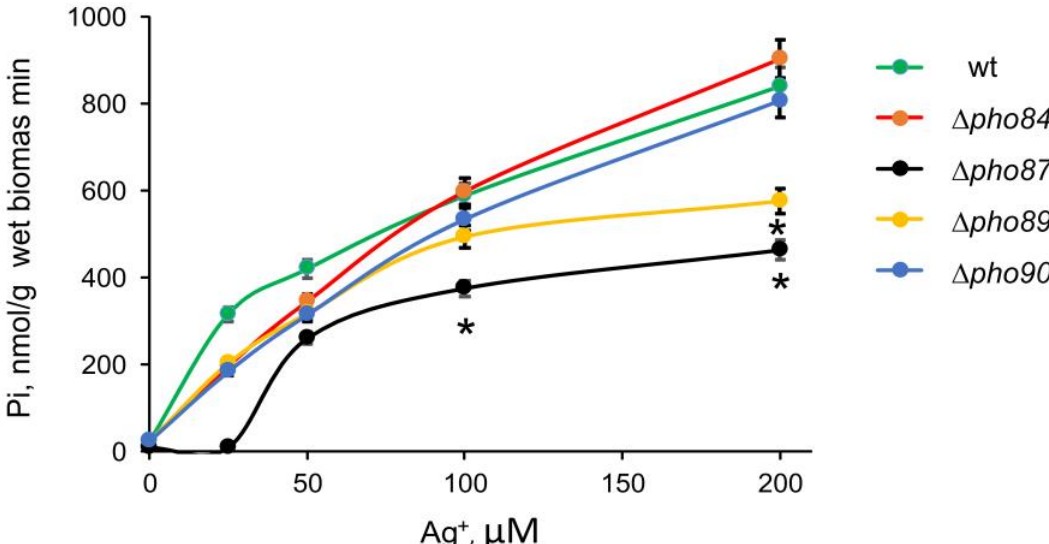

**Figure 2.** The effect of Ag$^+$ on the Pi release from the cells of wt and knockout *S. cerevisiae* strains. The cells were incubated at 30 °C for 30 min with AgNO$_3$ in water. The Pi release was measured as described in Methods. The experiments were performed in three replicates: the values denote mean and the whiskers denote s.d. * $p < 0.01$; no designation—n.s.; two-tailed Student's *T*-test against wt.

*3.3. The Effects of Mn$^{2+}$ on Yeast Growth and polyP Content*

In our previous studies, it was shown that the sensitivity of yeast cells to manganese ions depends on the polyP metabolism systems [45–47]. The strains overexpressing polyphosphatases Ppn1 and Ddp1 [45], as well as of knockout mutants lacking polyP synthetase Vtc4 [46] or polyphosphatase Ppn1 or polyphosphatase Ppn2 [47], have increased resistance to manganese. As for Pi transporters, only the effect of knockout mutation of *PHO84* is known: the enhanced resistance of Δ*pho84* knockout strain was explained by the fact that this transporter takes up manganese through the cytoplasmic membrane and the cells accumulate manganese in a lesser quantity compared to the wt strain [28].

The growth of all strains under study was suppressed by Mn$^{2+}$ in the concentration range used (Figure 3A). For this experiment, the cells were cultivated in immunoplate wells for 24 h. The sensitivities of the cells of wt and Δ*pho89* strains were similar (not illustrated). Therefore, in subsequent experiments, we did not test this mutant strain. The other three mutants differed from the wt strain in the sensitivity to Mn$^{2+}$. The growth of the Δ*pho84* and Δ*pho87* strains was suppressed to a lesser extent, and the growth of the Δ*pho90* strain was suppressed to a greater extent compared to the wt strain (Figure 3A).

To determine the content of manganese and polyP in the cells of the studied strains, cultivation was carried out in flasks in liquid medium, as described in Methods. When cultivated in flasks, a long lag phase was observed in the medium with 2 mM Mn$^{2+}$, and the stationary growth stage was reached significantly later than in control cultivation (24 h). The stationary stage in the medium with 2 mM Mn$^{2+}$ was reached in 4 days for the wt strain, in 3 days for the Δ*pho84* and Δ*pho87* strains, and in 6 days for the Δ*pho90* strain. Thus, the difference in sensitivity to manganese was expressed both when grown in plates and in flasks. Using the biomass samples obtained at the stationary growth stage, we determined the manganese content in the cells (Figure 3B). In the cells of the wt and Δ*pho84* strains, the manganese content was similar, in the cells of the Δ*pho87* strain it was significantly reduced, and in the cells of the Δ*pho90* strain it was increased as compared with the wt strain. The difference in sensitivity is likely related to the level of manganese accumulation.

Since polyP forms complexes with manganese, we determined the content of the short-chain fraction polyP1 and the long-chain fraction polyP2 (Figure 3C). Note that the differences in the polyP content between the strains under study were insignificant when the cells were cultivated in control YPD (Figure 3C). When the cultivation was performed in the presence of $Mn^{2+}$, the polyP content in the cells of all the studied mutant strains differed from the wt strain. The polyP1 content increased in strains wt and Δ*pho84* approximately equally compared to the controls. In the Δ*pho90* cells, the polyP1 content increased more than in the cells of the wt strain. The content of polyP2 in the cells of the Δ*pho84* and Δ*pho90* strains decreased more than in the cells of the wt strain. The polyP1 and polyP2 content in the cells of the Δ*pho87* strain was significantly decreased. The most indicative decrease in the polyP content was observed in the cells of the Δ*pho87* strain, which accumulated less manganese than other studied strains and was more resistant to the presence of this ion in the culture medium.

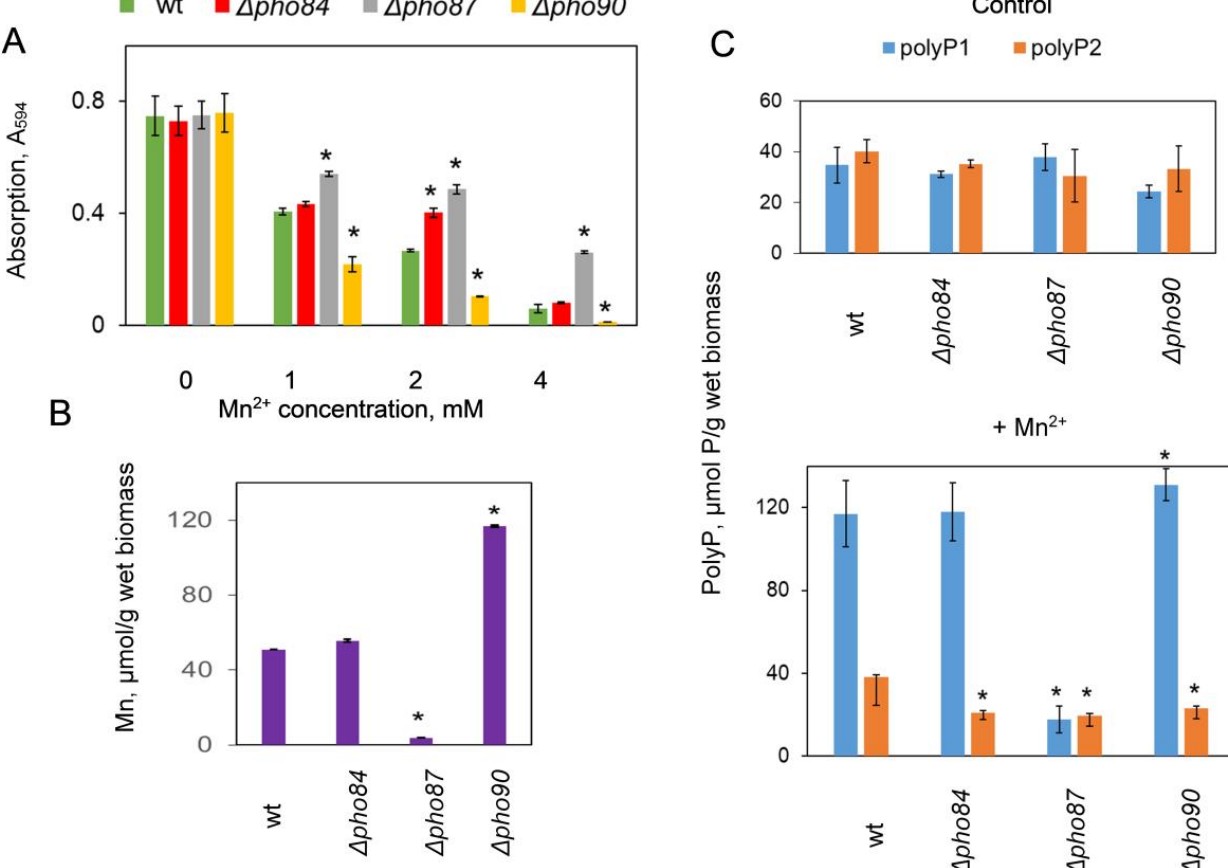

**Figure 3.** The effect of $Mn^{2+}$ on the growth, polyP, and manganese content in the cells of wt and mutant strains of *S. cerevisiae*. (**A**)—The effect of manganese on culture absorption; the cells were cultivated in immunoplates as described in Methods for 24 h. (**B**)—The content of manganese in the yeast biomass grown to the stationary stage in flasks with YPD supplemented with 2 mM $MnSO_4$. (**C**)—The content of polyP in the control cells (control) and in the cells cultivated in flasks to the stationary stage in the presence of 2 mM $MnSO_4$. The time of cultivation in the case of B and C was 4 days for the wt strain and 3, 3, and 6 days for the strains Δ*pho87*, Δ*pho84* and Δ*pho90*, respectively. The experiments were performed in four replicates; the values denote mean and the whiskers denote s.d. * $p < 0.01$; no designation—n.s.; two-tailed Student's *T*-test against wt.

## 4. Discussion

We demonstrated that the knockouts of genes encoding phosphate transporters of the cytoplasmic membrane of *S. cerevisiae* altered the sensitivity of cells to heavy metal

ions. These changes vary for different Pi carriers. However, we did not observe any link between the effects on resistance to heavy metal ions and affinity for Pi or Pi transport mechanisms. The $\Delta pho87$ and $\Delta pho89$ strains were more resistant to vanadate and silver ions. The $\Delta pho84$ and $\Delta pho87$ strains were more resistant to manganese, while the $\Delta pho90$ strain was less resistant.

The involvement of the *PHO84* gene in resistance to heavy metals is known. This transporter has been shown to be a low-affinity transporter of manganese [28], and therefore the knockout mutant accumulated half as much manganese and cobalt compared to the parental strain [48]. This explains its higher resistance to these toxic ions [48]. The higher resistance of the $\Delta pho84$ strain to manganese observed in our study (the lesser prolongation of the lag phase and the growth inhibition) is consistent with previously published data [28]. However, in our experiments, the $\Delta pho84$ mutant did not show a reliable difference from the wt strain in terms of the manganese accumulation. Note that the previous study [48] used the manganese concentration of 0.1 mM, while here we used a 20-times-higher concentration (2 mM $Mn^{2+}$). Apparently, this explains the difference in manganese accumulation. Interestingly, the strain with *PHO84* overexpression accumulated almost 5 times more $Mn^{2+}$ ions than the parent strain and was also more resistant to these ions. The authors [48] suggest that such overexpression leads to an increased flow of phosphate into the cell, and this ion, in turn, allows detoxification of excess manganese.

The most resistant to manganese, silver, and vanadate ions were the cells of the $\Delta pho87$ mutant. Although *PHO87* and *PHO90* are paralogs, their knockouts had distinct effects. For vanadate and silver exposure, $\Delta pho90$ did not differ from the parent strain, and in the case of manganese, $\Delta pho90$ sensitivity was significantly higher, in contrast to the more resistant $\Delta pho87$. We assume that the last effect is due to a decrease in the manganese accumulation and an increase in the manganese accumulation in the cells of the $\Delta pho87$ and $\Delta pho90$ strains, respectively. The different roles of these two transporters under phosphate limitation conditions have been established previously [19,26,32]. Pho87 and Pho90 have high similarities in the structure and contain the amino-terminal SPX domains sensing and binding inositol pyrophosphates (PP-InsPs). Inositol pyrophosphates serve as secondary messengers regulating the rate of Pi uptake, suppressing Pi efflux, and participating in the global regulation of the phosphate signaling pathway [49].

The question of interest is what structural features of these proteins lead to functional differences. The structure of the *S. cerevisiae* Pho90 (electron cryomicroscopy at 2.3 and 3.1 Å resolution) was established both in the absence and presence of Pi [50]. For Pho87, there are no such data yet. According to UniProt [51], the amino acid sequences of these proteins are highly homologous in the region encoding the transmembrane domains, responsible for phosphate transfer. In turn, while the arrangement of alpha-helices of the SPX domains is also highly similar, the connecting unstructured loops differ significantly between Pho87 and Pho90. Furthermore, according to the ProteomeXchange [52] data on post-translational modifications, these loops of Pho87 carry many phosphorylated serines, and the resulting negative charge might be affecting the SPX domain binding affinity to PP-InsPs.

The reason why the cells of these knockout strains differ in their ability to accumulate manganese may be the involvement of these transporters in the regulatory pathways of manganese uptake and export. The *S. cerevisiae* $Mn^{2+}$ transporters Smf1 and Smf2 (from the Nramp family of metal transporters) and the phosphate transporter Pho84 have different roles in $Mn^{2+}$ homeostasis [17,24]. Smf2 is a high-affinity plasma membrane transporter that takes up extracellular $Mn^{2+}$ under conditions of $Mn^{2+}$ deficiency. Smf2 is localized in Golgi-like vesicles and is a low-affinity $Mn^{2+}$ transporter [17,24]. When $Mn^{2+}$ concentrations are stable or excessive, Smf1 and Smf2 are ubiquitinated and transported to

the vacuole for degradation [17]. Using Δ*pho84* mutants, it was shown that Mn$^{2+}$ uptake is most often impaired upon deletion of *PHO84* [17]. Transcription of *PHO84* is regulated by the transcription factor Pho4. Note that a decrease in the *PHO84* gene expression was observed in a strain overexpressing the Ppn1 polyphosphatase, which exhibited high resistance to manganese [53]. Taken together, these data indicate multiple interactions between manganese and Pi homeostasis, and a deeper understanding of how phosphate transporters Pho87 and Pho90 are involved in this process apparently requires focused transcriptomic and biochemical studies.

## 5. Conclusions

We demonstrated that the knockouts of the genes encoding plasma membrane Pi transporters changed the resistance of *S. cerevisiae* cells to toxic concentrations of manganese, silver, and vanadate ions. The knockout of *PHO87* encoding plasma membrane Pi transporter of *S. cerevisiae* led to increased resistance to silver, vanadate, and manganese ions. On the contrary, the knockout mutation in the *PHO90* gene, even though the genes are paralogs, decreased resistance to manganese ions and showed no effect on the resistance to vanadate or silver. Strains with altered expression of phosphate transporters may be of interest for use in bioremediation or monitoring of heavy metal contamination. In particular, although the Δ*pho90* cells grow slowly in the presence of manganese, the biomass accumulates significantly more manganese, which can be of interest for the practical removal of this ion from aquatic environments. Strains with increased resistance to manganese may be of interest for biomonitoring, since they are able to survive at higher concentrations of the pollutant.

**Author Contributions:** Conceptualization, T.K.; methodology, L.L. and L.T.; validation, L.R. and L.L.; investigation, L.L., L.R., L.T. and V.O.; data curation, L.L., L.R., L.T. and V.O.; writing—original draft preparation, T.K.; writing—review and editing, T.K. All authors have read and agreed to the published version of the manuscript.

**Funding:** This research received no external funding.

**Institutional Review Board Statement:** Not applicable.

**Data Availability Statement:** The original contributions presented in this study are included in the article. Further inquiries can be directed to the corresponding author.

**Conflicts of Interest:** The authors declare no conflicts of interest.

## Abbreviations

The following abbreviations are used in this manuscript:

| | |
|---|---|
| Pi | Orthophosphate |
| PolyP | Inorganic polyphosphate |

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
