# Peer review of "Knockouts of Yeast Plasma Membrane Phosphate Transporters Alter Resistance to Heavy Metals"

_2673-8007, doi:10.3390/applmicrobiol5040109_

Round 1
Reviewer 1 Report
Comments and Suggestions for Authors
In this manuscript, Ledova and co-authors present an interesting study regarding the role of genes encoding plasma membrane phosphate transporters from Saccharomyces cerevisiae in cell tolerance to heavy metal stress. The study is timely and may be of technological relevance, offering clues on the mechanisms of metal absorption and detoxification of heavy metal pollutants.
The manuscript is well written, following a coherent and logical thread. There are some issues that the authors need to address before the manuscript can be considered for publication.
- The authors normalized Mn2+, polyP, etc. to gram of wet biomass, which seems a bit un-precise. The authors are advised to explain in more detail the procedure used to obtain the wet biomass considered.
- Line 109: correct 200 mL to 200 microliters.
- Line 119: replace “After them” with “After that”.
- Figure 3A: Introduce title for OX axis, i.e. Mn2+ concentration (mM).
Author Response
Thank you very much for reviewing and useful comments. We have revised the text in accordance with your comments.
The authors normalized Mn2+, polyP, etc. to gram of wet biomass, which seems a bit un-precise. The authors are advised to explain in more detail the procedure used to obtain the wet biomass considered.
- To compare the polyP content, we used the quantities of polyP as Pi normalized to sample masses (µmol P/ g of raw biomass). The raw biomass samples were standardized using the same centrifugation conditions, 5000 g for 15 min as in our previous works.
Line 109: correct 200 mL to 200 microliters.
- It's done
Line 119: replace “After them” with “After that”.
- It's done
Figure 3A: Introduce title for OX axis, i.e. Mn2+ concentration (mM).
- It's done
Sincerely,
Tatiana Kulakovskaya, on behalf of all co-authors
Reviewer 2 Report
Comments and Suggestions for Authors
Reviewer Comments
General Assessment
The manuscript investigates the role of plasma membrane phosphate transporters (PHO84, PHO87, PHO89, PHO90) in S. cerevisiae resistance to heavy metal ions (manganese, silver, vanadate). The topic is relevant to microbial physiology and environmental biotechnology, especially in the context of heavy metal bioremediation. The study is clearly presented and includes appropriate experiments, but there are several issues regarding clarity, depth of discussion, methodology description, and positioning within the broader literature.
Major Comments
- Novelty and Positioning
- The manuscript needs a clearer emphasis on novelty. While the roles of PHO84 are documented, the specific new findings about PHO87 and PHO90 should be more explicitly contrasted with prior work.
- The introduction should highlight why these paralogs (PHO87 vs. PHO90) might behave differently, based on structural or regulatory domains (SPX domain), before presenting results.
- Methodology Details
- The methods section lacks sufficient detail for reproducibility in some parts:
- Concentration ranges of Ag+ used are not clearly listed in the text.
- The description of statistical analysis is vague; please specify n values per experiment and whether data normality was checked before applying Student’s t-test.
- The polyphosphate assay method relies on earlier references but should provide at least one or two sentences summarizing critical steps.
- Data Presentation
- Figures are described, but the actual figures (1–3) should be clearer and self-contained with complete legends (including media composition, concentrations tested, and biological replicates).
- The statistical significance indicators (, n.s.) are explained but inconsistently applied across figure captions. Standardize.
- Discussion Depth
- The discussion is largely descriptive. It should integrate mechanistic insights:
- How do PHO87 and PHO90 transport activities differ at the molecular level, and how might this explain the contrasting phenotypes?
- Could cross-talk with other metal transporters (e.g., Smf1/Smf2 manganese transporters) explain the observed accumulation patterns?
- The possible link between altered phosphate homeostasis and polyphosphate metabolism in Δpho87 vs Δpho90 needs more mechanistic discussion.
- Biotechnological Relevance
- The conclusion mentions bioremediation, but practical implications are not well developed. Please elaborate:
- Could Δpho87 strains be used as biosorbents due to reduced metal accumulation but higher tolerance?
- Are there limitations (e.g., stability, scalability, safety concerns)?
Minor Comments
- Language and Grammar
- Several sentences are grammatically awkward (e.g., line 34 “Pollution of water and soil with heavy metal ions an important issue…” → “is an important issue”). A thorough language edit is recommended.
- Consistency: use either “wild-type strain” or “wt strain” throughout.
- References
- Many references are recent (2023–2025), which is good. However, some classical references on yeast phosphate transport and metal homeostasis are missing. Consider adding more background on manganese transporters (Smf1, Smf2) and the PHO pathway.
- Abbreviations
- Define all abbreviations at first use in the text (e.g., Pi, polyP).
- Figures
- Ensure that axis labels, units, and strain names are readable in all figures.
- Author Contributions Section
- This section has typos and incomplete names (“Y.Y.”, “L..”). Please correct.
-
- Several sentences are grammatically awkward (e.g., line 34 “Pollution of water and soil with heavy metal ions an important issue…” → “is an important issue”). A thorough language edit is recommended.
- Consistency: use either “wild-type strain” or “wt strain” throughout.
Author Response
Thank you very much for reviewing and useful comments. We have revised the text in accordance with your comments.
“The manuscript needs a clearer emphasis on novelty. While the roles of PHO84 are documented, the specific new findings about PHO87 and PHO90 should be more explicitly contrasted with prior work.
The introduction should highlight why these paralogs (PHO87 vs. PHO90) might behave differently, based on structural or regulatory domains (SPX domain), before presenting results.”
We added in the Introduction:
- Of particular interest are Pho87 and Pho90: they are encoded by paralogous genes, but exhibit functional differences [26, 32]. This has been clearly shown using the knockout of PHO4, which encodes the transcription factor Pho4 that activates the expression of genes of phosphate starvation response under Pi limitation. In Δpho4 cells, rapid res-toration of Pi transport and growth at phosphate limitation is provided by the Pho87 transporter, but not by Pho90 [32]. The conditions of encocytosis into the vacuole are also different for these proteins [32]. Until now, there has been no data reported re-garding the participation of Pho87, Pho89, or Pho90 in the heavy metal resistance mechanisms.
- The aim of this study was to compare S. cerevisiae knockouts of plasma membrane Pi transporters PHO87, PHO90, and PHO89 in terms of resistance to silver, vanadate, and manganese.
The methods section lacks sufficient detail for reproducibility in some parts:
Concentration ranges of Ag+ used are not clearly listed in the text. The description of statistical analysis is vague; please specify n values per experiment and whether data normality was checked before applying Student’s t-test.
The polyphosphate assay method relies on earlier references but should provide at least one or two sentences summarizing critical steps.
- We indicated the concentration range of Ag in the text.
- The description of statistics was changed:
The experiments were performed in 3-4 replicas, the number of replicas is indicated in the figure captions. All figures show average values from the indicated number of experiments, the whiskers denote s.d. Statistical analysis was assessed with the two-tailed Student’s T-test using standard Excell software and was performed for data for mutant strains studied against data for wt strain.
We have expanded the description of the polyP extraction method:
The cells were separated from the medium by centrifugation at 5000 g for 15 min and washed two times with distilled water under the same centrifugation conditions. Wet biomass samples, 150 mg each, were frozen, stored at –20°С, and used for polyP extraction and assay. Pi, acid-soluble polyP fraction (polyP1) and acid-insoluble polyP fraction (polyP2) were extracted at 0°С as described previously. The acid-soluble polyP1 was extracted from biomass samples by the treatment with 0.5 M HClO4 at 0° for 20 min under continuous stirring. After that, the samples were centrifuged at 7000 g for 3 min and supernatants collected. The extraction was repeated twice. The combined supernatants contained acid-soluble polyP1. The content of Pi before hydrolysis and after polyP hydrolysis in 1 M HCl for 20 min at 90 ºC [27] was determined. The difference between these values was the content of acid-soluble polyP1 expressed as Pi [27]. The remaining biomass was treated with 1M HClO4 at 100° for 10 min, After cooling, the samples were centrifuged at 7000 g for 3 min and the content of acid-insoluble polyP2 s was assayed by the Pi quantification. Pi was quantified by a malachite green colorimetric assay with a microplate photometer (MZ Sapphire, Russia). To compare the polyP content, we used the quantities of polyP as Pi normalized to sample masses (µmol P/ g of raw biomass). The raw biomass samples were standardized using the same centrifugation conditions, 5000 g for 15 min as in our previous works .
Data Presentation
Figures are described, but the actual figures (1–3) should be clearer and self-contained with complete legends (including media composition, concentrations tested, and biological replicates).
The statistical significance indicators (, n.s.) are explained but inconsistently applied across figure captions. Standardize.
-We corrected the figures and figure captions
Discussion Depth
The discussion is largely descriptive. It should integrate mechanistic insights:
How do PHO87 and PHO90 transport activities differ at the molecular level, and how might this explain the contrasting phenotypes?
Could cross-talk with other metal transporters (e.g., Smf1/Smf2 manganese transporters) explain the observed accumulation patterns?
The possible link between altered phosphate homeostasis and polyphosphate metabolism in Δpho87 vs Δpho90 needs more mechanistic discussion.
We rewrote the Discussion section.
Biotechnological Relevance
The conclusion mentions bioremediation, but practical implications are not well developed. Please elaborate:
Could Δpho87 strains be used as biosorbents due to reduced metal accumulation but higher tolerance?
Are there limitations (e.g., stability, scalability, safety concerns)?
We rewrote the Conclusion section
Minor Comments
Language and Grammar
Several sentences are grammatically awkward (e.g., line 34 “Pollution
of water and soil with heavy metal ions an important issue…” → “is an
important issue”). A thorough language edit is recommended.
Consistency: use either “wild-type strain” or “wt strain” throughout.
-it is corrected
References
Many references are recent (2023–2025), which is good. However, some
classical references on yeast phosphate transport and metal
homeostasis are missing. Consider adding more background on manganese
transporters (Smf1, Smf2) and the PHO pathway.
We added references about PHO pathway and manganese transporters to Discussion and Introduction sections
Abbreviations
Define all abbreviations at first use in the text (e.g., Pi, polyP).
- It is done
Figures
Ensure that axis labels, units, and strain names are readable in all figures.
- It is done
Author Contributions Section
This section has typos and incomplete names (“Y.Y.”, “L..”). Please correct.
Several sentences are grammatically awkward (e.g., line 34 “Pollution
of water and soil with heavy metal ions an important issue…” → “is an
important issue”). A thorough language edit is recommended.
- We have done language editing
Consistency: use either “wild-type strain” or “wt strain” throughout.
- It is done.
Sincerely,
Tatiana Kulakovskaya, on behalf of all co-authors
Round 2
Reviewer 1 Report
Comments and Suggestions for Authors
The authors responded to reviewer's concerns.
Author Response
Dear Reviewer 1,
Thank you very much for reviewing of our manuscript.
Sincerely,
Tatiana Kulakovskaya
Reviewer 2 Report
Comments and Suggestions for Authors
Minor Comments
- Title: Consider rephrasing for conciseness and impact. Example:
“Knockouts of yeast plasma membrane phosphate transporters alter resistance to heavy metals.”
- Abstract: The sentence “Δpho87 cells accumulated 10 times less manganese compared to wild type” is strong. Please specify the experimental condition to avoid overgeneralization.
- Methods:
- Specify the source and purity of AgNO₃ and Na₃VO₄.
- Clarify whether Mn content measurements were normalized per dry biomass or wet biomass (currently ambiguous).
- Grammar/Language:
-Several sentences are long and complex. Editing for clarity and conciseness is recommended.
-Example: Line 87–89: “The expression of the PHO87 and PHO90 genes is independent of Pi concentration in the medium, while transcription of PHO84 and PHO89 is induced under Pi limitation.” → Could be shortened for readability.
- References:
-Some references are outdated; please ensure the most recent relevant literature on PHO transporters and heavy metal detoxification is included.
-Check formatting consistency (e.g., spacing, punctuation in references 9, 10, 31).
Author Response
Dear Reviewer 2,
Thank you very much for the reviewing of our manuscript ID: applmicrobiol-3865893.
- Title: Consider rephrasing for conciseness and impact. Example:
“Knockouts of yeast plasma membrane phosphate transporters alter resistance to heavy metals.”
- We rephrased the title:
Knockouts in the genes encoding yeast plasma membrane phosphate transporters alter resistance to heavy metals ions.
- Abstract: The sentence “Δpho87 cells accumulated 10 times less manganese compared to wild type” is strong. Please specify the experimental condition to avoid overgeneralization.
- We specify the conditions - The Δpho87 cells accumulated 10 times less manganese compared to the wild-type cells, while the Δpho90 cells accumulated two times more compared to the wild-type cells, when grown in YPD with 2 mM MnSO4
- Methods:
- Specify the source and purity of AgNO₃ and Na₃VO₄.
- We indicated the source and purity of there salts in Methods
- Clarify whether Mn content measurements were normalized per dry biomass or wet biomass (currently ambiguous).
- We changed the term “raw biomass” for “wet biomass” in the manuscript.
- Grammar/Language:
-Several sentences are long and complex. Editing for clarity and conciseness is recommended.
We edited some long sentences.
-Example: Line 87–89: “The expression of the PHO87 and PHO90 genes is independent of Pi concentration in the medium, while transcription of PHO84 and PHO89 is induced under Pi limitation.” → Could be shortened for readability.
- References:
Some references are outdated; please ensure the most recent relevant literature on PHO transporters and heavy metal detoxification is included.
- According to recommendations of Reviewer “Many references are recent (2023–2025), which is good. However, some classical references on yeast phosphate transport and metal
homeostasis are missing. Consider adding more background on manganese transporters (Smf1, Smf2) and the PHO pathway”
we added several classical references, for example
Mouillon, J.M.; Persson, B.L. New aspects on phosphate sensing and signalling in Saccharomyces cerevisiae. FEMS Yeast Res 2006, 6: 171–176. https://doi.org/10.1111/j.1567-1364.2006.00036.x.
Reddi, A.R.; Jensen, L.T.; Culotta, V.C. Manganese homeostasis in Saccharomyces cerevisiae. Chem Rev 2009, 109(10) 4722-4732. doi: 10.1021/cr900031u. PMID: 19705825; PMCID: PMC3010240.
Auesukaree, C.; Homma, T.; Kaneko, Y.; Harashima, S. Transcriptional regulation of phosphate-responsive genes in low-affinity phosphate-transporter-defective mutants in Saccharomyces cerevisiae. Biochem. Biophys. Res. Commun. 2003, 306, 843–850.
Bun-Ya, M.; Nishimura, M.; Harashima, S.; Oshima, Y. The PHO84 Gene of Saccharomyces cerevisiae encodes an inorganic
phosphate transporter. Mol. Cell. Biol. 1991, 11, 3229–3238.
and most resent references
Bazzicalupo, A.L.; Kahn, P.C.; Ao, E.l Campbell, J.; Otto, S.P. Evolution of cross-tolerance to metals in yeast. Proc Natl Acad Sci U S A 2025, 122(37), e2505337122. doi: 10.1073/pnas.2505337122. PMID: 40928868.
Eskes, E.; Deprez, M.A.; Wilms, T.; Winderickx, J. pH homeostasis in yeast; the phosphate perspective. Curr Genet 2018, 64,155–161. doi: 10.1007/s00294-017-0743-2.
Check formatting consistency (e.g., spacing, punctuation in references 9, 10, 31).
- It was done
Sincerely,
Tatiana Kulakovskaya